# Genetic Associations of *ITGB3*, *FGG*, *GP1BA*, *PECAM1*, and *PEAR1* Polymorphisms and the Platelet Activation Pathway with Recurrent Pregnancy Loss in the Korean Population

**DOI:** 10.3390/ijms26157505

**Published:** 2025-08-03

**Authors:** Eun Ju Ko, Eun Hee Ahn, Hyeon Woo Park, Jae Hyun Lee, Da Hwan Kim, Young Ran Kim, Ji Hyang Kim, Nam Keun Kim

**Affiliations:** 1Division of Life Science, College of Life Science, CHA University, Seongnam 13488, Republic of Korea; ejko05@naver.com (E.J.K.); aabb1114@naver.com (H.W.P.); athe7a@naver.com (J.H.L.); soda00831kr@naver.com (D.H.K.); 2Department of Life Science, Graduate School, CHA University, Seongnoam 13488, Republic of Korea; 3Department of Obstetrics and Gynecology, CHA Bundang Medical Center, School of Medicine, CHA University, Seongnam 13596, Republic of Korea; bestob@chamc.co.kr (E.H.A.); happyiran@chamc.co.kr (Y.R.K.)

**Keywords:** recurrent pregnancy loss, platelet activation, single-nucleotide polymorphism, genotype combination, allele combination

## Abstract

Recurrent pregnancy loss (RPL) is defined as the occurrence of two or more pregnancy losses before 20 weeks of gestation. RPL is a common medical condition among reproductive-age women, with approximately 23 million cases reported annually worldwide. Up to 5% of pregnant women may experience two or more consecutive pregnancy losses. Previous studies have investigated risk factors for RPL, including maternal age, uterine pathology, genetic anomalies, infectious agents, endocrine disorders, thrombophilia, and immune dysfunction. However, RPL is a disease caused by a complex interaction of genetic factors, environmental factors (e.g., diet, lifestyle, and stress), epigenetic factors, and the immune system. In addition, due to the lack of research on genetics research related to RPL, the etiology remains unclear in up to 50% of cases. Platelets play a critical role in pregnancy maintenance. This study examined the associations of platelet receptor and ligand gene variants, including integrin subunit beta 3 (*ITGB3*) rs2317676 A > G, rs3809865 A > T; fibrinogen gamma chain (*FGG*) rs1049636 T > C, rs2066865 T > C; glycoprotein 1b subunit alpha (*GP1BA*) rs2243093 T > C, rs6065 C > T; platelet endothelial cell adhesion molecule 1 (*PECAM1*) rs2812 C > T; and platelet endothelial aggregation receptor 1 (*PEAR1*) rs822442 C > A, rs12137505 G > A, with RPL prevalence. In total, 389 RPL patients and 375 healthy controls (all Korean women) were enrolled. Genotyping of each single nucleotide polymorphism was performed using polymerase chain reaction–restriction fragment length polymorphism and the TaqMan genotyping assay. All samples were collected with approval from the Institutional Review Board at Bundang CHA Medical Center. The *ITGB3* rs3809865 A > T genotype was strongly associated with RPL prevalence (pregnancy loss [PL] ≥ 2: adjusted odds ratio [AOR] = 2.505, 95% confidence interval [CI] = 1.262–4.969, *p* = 0.009; PL ≥ 3: AOR = 3.255, 95% CI = 1.551–6.830, *p* = 0.002; PL ≥ 4: AOR = 3.613, 95% CI = 1.403–9.307, *p* = 0.008). The *FGG* rs1049636 T > C polymorphism was associated with a decreased risk in women who had three or more pregnancy losses (PL ≥ 3: AOR = 0.673, 95% CI = 0.460–0.987, *p* = 0.043; PL ≥ 4: AOR = 0.556, 95% CI = 0.310–0.997, *p* = 0.049). These findings indicate significant associations of the *ITGB3* rs3809865 A > T and *FGG* rs1049636 T > C polymorphisms with RPL, suggesting that platelet function influences RPL in Korean women.

## 1. Introduction

South Korea’s birth rate, reaching its lowest level since birth statistics were first recorded, has become a critical social issue. In 2023, the country’s total fertility rate was 0.72, which is substantially lower than the Organization for Economic Cooperation and Development average of 1.58 [1]. Additionally, 17.2% of married women experienced infertility, while 31.9% of married women over the age of 35 at first marriage reported infertility [2]. These trends have continued to increase annually. Infertility causes physical discomfort and contributes to psychological distress, including anxiety and depression; it also constitutes an economic burden due to the cost of infertility treatment [3].

Recurrent pregnancy loss (RPL) is a cause of infertility. Pregnancy loss generally refers to the loss of an intrauterine pregnancy before 20 weeks of gestation and occurs sporadically in most cases. It is estimated that fewer than 5% of couples who experience pregnancy loss have two consecutive losses, and only 1% of couples experience three or more losses. However, mothers who have experienced one pregnancy loss exhibit an increased risk of subsequent losses. The risk of pregnancy loss after a single previous loss is 12–20%; this risk increases to 29% after two losses and 36% after three losses [4]. Risk factors for RPL include maternal age, uterine pathology, genetic abnormalities, infectious agents, thrombophilia, and immune dysfunction. Advanced maternal age is associated with an increased risk of chromosomal abnormalities, as well as a decline in uterine and hormonal function. Pathological abnormalities within the uterus may reduce embryo implantation rates or cause insufficient space for fetal growth. In addition, certain genetic mutations (e.g., chromosomal abnormalities, single-gene disorders, and confined placental mosaicism) may lead to fetal growth restriction or placental dysfunction [5]. Platelets, which are the focus of this study, are involved in both coagulation and the immune system. Platelets release coagulation factors and serve as a scaffold during thrombus formation. Also, platelets interact directly with immune cells or modulate their function by releasing soluble factors [6]. Immune cells in the uterus contribute to embryo implantation, angiogenesis, and placental development. A prothrombotic state during pregnancy may result in venous thromboembolism at the implantation site or placental vessels. Immune dysregulation may lead to the increased infiltration of immune cells into the placenta or induce thrombus formation within the chorionic villi or placenta [7]. Thus, platelet dysfunction may contribute to thrombosis or immune dysregulation during pregnancy.

Platelets, which are the smallest blood components produced by megakaryocytes, play critical roles in thrombosis and hemostasis, as well as in immune responses and vascular integrity [8]. Additionally, they contribute to ovarian function, including folliculogenesis and oocyte maturation, along with endometrial development, implantation, placentation, and embryonic growth. Platelets are also implicated in the pathophysiology of various reproductive disorders, including ovarian hyperstimulation syndrome; preeclampsia; pregnancy-induced hypertension; hemolysis and low platelet level syndrome; and intrauterine growth restriction [9]. The cytokines and growth factors present in platelets have been known to promote wound healing and the regeneration of damaged tissues. Platelet-rich plasma (PRP), which is a platelet concentrate derived from whole blood, has emerged as a promising therapeutic approach for infertility treatment [10]. Studies have demonstrated that the intra-ovarian injection of PRP promotes oocyte maturation and regulates angiogenesis, indicating its potential as a treatment for premature ovarian insufficiency [11]. Furthermore, PRP can stimulate endometrial growth and enhance endometrial receptivity. In cases of recurrent implantation failure, the intrauterine injection of autologous PRP 24 to 48 h before embryo transfer has been associated with successful implantation and live birth outcomes [12].

Transmembrane receptors on the platelet surface mediate platelet adhesion (glycoprotein [GP]Ib-IX-V complex), aggregation (integrin αIIbβ3), inflammatory responses, and immune regulation (e.g., GPVI and C-type lectin-like receptor 2 [CLEC-2]) [13]. In this study, we selected five genes (*ITGB3*, *FGG*, *GP1BA*, *PECAM1*, and *PEAR1*) related to platelet activation for analysis (Figure 1). The *ITGB3* gene encodes integrin subunit beta 3 (GPIIIa, CD61), which is a component of the platelet receptor αIIbβ3 complex. This complex binds fibrinogen and von Willebrand factor to mediate platelet aggregation. The *FGG* gene encodes the gamma chain of fibrinogen. Fibrinogen is converted to fibrin to form a blood clot, or it binds to the αIIbβ3 complex on the platelet surface to activate platelets. The *GP1BA* gene encodes glycoprotein Ib alpha chain (GPIB, CD42b), which is a component of the GPIb-IX-V complex on the platelet surface. The GPIb-IX-V complex binds von Willebrand factor to mediate platelet adhesion [14]. The *PECAM1* gene is expressed in vascular endothelial cells and platelets; its protein product (platelet endothelial cell adhesion molecule 1) regulates platelet–vessel wall interactions, vascular permeability, and inflammatory inhibition [15]. Finally, the *PEAR1* gene encodes platelet endothelial aggregation receptor 1, which is a platelet membrane protein that is essential for platelet activation and aggregation [16].

More than half of pregnancy losses are attributed to genetic defects, such as an abnormal number of chromosomes in the embryo or genetic rearrangements. However, more complex and extensive genetic defects beyond chromosomal abnormalities may also contribute to pregnancy loss [17]. Such genetic causes include single-nucleotide polymorphisms (SNPs), copy number variants, and insertions/deletions (i.e., indels). An SNP involves the substitution of one nucleotide for another. SNPs are represented by a reference allele, a variant allele, and a SNP number. For example, in the case of *ITGB3* rs2317676A > G, *ITGB3* is the gene where the SNP is located, > is a symbol indicating a substitution, adenine (A) is the reference allele found in the general population, and guanine (G) represents the variant allele. SNPs can lead to the production of proteins with altered functions or modified protein expression levels, thereby contributing to disease pathology. In this study, we selected SNPs from genes associated with platelet activation receptors and their ligands (*ITGB3*, *FGG*, *GP1BA*, *PECAM1*, and *PEAR1*, as mentioned above). These SNPs were chosen based on previously reported clinical studies, an appropriate minor allele frequency (≥0.05), and their potential impact on gene expression. Specifically, SNPs in promoter regions, missense variants, and those in 5′ and 3′ untranslated regions (UTRs) were considered. The *ITGB3* rs2317676 A > G, rs3809865 A > T, *FGG* rs1049636 T > C, and *PECAM1* rs2818 C > T variants are in the 3′ UTR, which is a region involved in miRNA binding, mRNA stability, and secondary structure formation [18]. The *FGG* rs2066865 T > C variant is located 500 base pairs downstream in a GT-rich sequence containing a putative binding site for cleavage stimulation factor, which is involved in the efficient cleavage and polyadenylation of pre-mRNA [19]. The *GP1BA* rs6065 C > T, *PEAR1* rs822442 C > A, and *PEAR1* rs12137505 C > G variants are missense mutations in exon regions that alter amino acid sequences; *GP1BA* rs2243093 T > C is in the 5′ UTR, where transcription factor binding sites and ribosome binding sites are present (Appendix A). RPL is influenced by the individual and cumulative effects of environmental and genetic factors [20]. In this study, we analyzed the cumulative effects of environmental factors (e.g., hormone levels, complete blood counts, and blood biochemistry findings) and genetic factors. Additionally, these SNPs have been associated with multiple vascular diseases related to hemostasis, which play a critical role in pregnancy maintenance. Therefore, we assessed relationships between platelet-related gene SNPs and clinical factors with RPL prevalence to identify potential diagnostic biomarkers for RPL and possible therapeutic targets.

## 2. Results

### 2.1. Clinical Characteristics of the Study Participants

We analyzed differences between control and RPL patient groups to determine the clinical characteristics of the study population. Table 1 presents the clinical variables of 389 RPL patients and 375 controls. After age matching, the mean age of control and RPL participants was 32.85 ± 4.18 years and 33.42 + 4.35 years, respectively; no statistically significant difference in age distribution was evident between the control and RPL groups (*p* = 0.275). Among female hormone levels, follicle-stimulating hormone levels were lower in the RPL patient group, whereas estradiol and luteinizing hormone levels were higher. Thyroid-related markers, including TSH and anti-TPO, were significantly elevated in RPL patients (all *p* < 0.005). The blood coagulation-related factors prothrombin time and activated partial thromboplastin time were significantly higher in the patient group (both *p* < 0.001). Significant differences were observed between the two groups in terms of white blood cell count, red blood cell count, hemoglobin level, red blood cell distribution width, platelet count, platelet distribution width, mean platelet volume, segmented neutrophil count, lymphocyte count, eosinophil count, and basophil count (all *p* < 0.05). No significant differences in other factors were identified between the RPL and control groups.

### 2.2. Genotype and Allele Frequencies of the ITGB3, FGG, GP1BA, PECAM1, and PEAR1 Gene Polymorphisms

We compared genotype and allele frequencies between the two groups to assess the relationships of specific SNPs (*ITGB3* rs2317676 A > G, rs3809865 A > T; *FGG* rs1049636 T > C, rs2066865 T > C; *GP1BA* rs2243093 T > C, rs6065 C > T; *PECAM1* rs2812 C > T; and *PEAR1* rs822442 C > A, rs12137505 G > A) with RPL risk and to identify genetic variants associated with the condition. Logistic regression analysis was performed to compare risk between the groups. AORs were calculated via logistic regression, with age as a covariate. Genotype frequencies in both the control and RPL groups met Hardy–Weinberg equilibrium criteria (all *p* > 0.05). The *ITGB3* rs3809865 A > T polymorphism showed a significant difference in the TT genotype and recessive models in the RPL group compared with controls (AA vs. TT: AOR = 2.505, 95% CI = 1.262–4.969, *p* = 0.009; AA + AT vs. TT: AOR = 2.302, 95% CI = 1.176–4.508, *p* = 0.015; Table 2). We also investigated associations of the nine polymorphisms with pregnancy loss number (PL) (Table 2). Among patients with a higher number of pregnancy losses, the *ITGB3* rs3809865 TT genotype and recessive models were associated with increased RPL risk (TT: PL ≥ 3 group: AOR = 3.255, 95% CI = 1.551–6.830, *p* = 0.002; PL ≥ 4: AOR = 3.613, 95% CI = 1.406–9.307, *p* = 0.008; AA + AT vs. TT: PL ≥ 3 group: AOR = 3.050, 95% CI = 1.479–6.292, *p* = 0.003; PL ≥ 4: AOR = 3.301, 95% CI = 1.316–8.280, *p* = 0.011; Table 2). The *FGG* rs1049636 T > C polymorphism was significantly associated with decreased susceptibility to RPL in patients with three or more pregnancy losses (TC: PL ≥ 3 group: AOR = 0.673, 95% CI = 0.460–0.987, *p* = 0.043; PL ≥ 4: AOR = 0.556, 95% CI = 0.310–0.997, *p* = 0.049; TT + TC vs. CC: PL ≥ 3 group: AOR = 0.669, 95% CI = 0.463–0.968, *p* = 0.033; Table 2). Additionally, among patients with three or more pregnancy losses, the *ITGB3* rs3809865 TT genotype remained significantly associated after false discovery rate (*FDR*-) correction (*FDR*-*p* = 0.018 for AA vs. TT; *FDR*-*p* = 0.027 for AA + AT vs. TT). No other associations remained statistically significant (*p* > 0.05; Appendix A). This analysis identified a strong link between the *ITGB3* rs3809865 A > T TT genotype and increased RPL risk, suggesting a genetic contribution of this variant to RPL pathogenesis.

### 2.3. Allele Combination and Haplotype Analysis of the ITGB3, FGG, GP1BA, PECAM1, and PEAR1 Gene Polymorphisms

We analyzed allele combinations for the nine SNPs to explore potential interactions across polymorphic sites that may exert a combinatorial effect on RPL risk (Table 3, Appendix A). Among the allele combinations, A.T.T.C.T.C.C.C.G, which includes the mutant T allele from *ITGB3* rs3809865 A > T and the mutant C allele from *FGG* rs2066865 T > C, showed the strongest association with increased RPL risk (OR = 27.430, *FDR-p* = 0.023). Conversely, the combinations A.A.T.T.T.C.T.A.A, A.A.T.C.C.C.C.A.A, A.T.T.T.T.C.C.C.G, and A.T.C.C.C.C.C.C.G were associated with decreased RPL risk (OR = 0.022, *FDR-p* = 0.008; OR = 0.104, *FDR-p* = 0.008; OR = 0.239, *FDR-p* = 0.010; OR = 0.041, *FDR-p* = 0.030, respectively).

To reduce the dimensionality of the nine polymorphisms and identify meaningful allele combinations, we performed MDR analysis. Accordingly, *ITGB3* rs2317676 A > G/*ITGB3* rs3809865 A > T/*FGG* rs1049636 T > C/*PEAR1* rs12137505 G > A was selected in four allele combinations; *ITGB3* rs2317676 A > G/*ITGB3* rs3809865 A > T/*GP1BA* rs6065 C > T was selected in three combinations; and *ITGB3* rs3809865 A > T/*GP1BA* rs6065 C > T was selected in two combinations (Table 3). In the four-locus allele combination groups (*ITGB3* rs2317676 A > G/*ITGB3* rs3809865 A > T/*FGG* rs1049636 T > C/*PEAR1* rs12137505 G > A), the A.T.C.A combination was associated with decreased RPL risk (OR = 0.388, *FDR-p* = 0.045); the risk significantly increased (OR = 17.820) in the G.T.C.A combination, which contained all mutant alleles (*FDR-p* = 0.045). In the three-locus allele combination group (*ITGB3* rs2317676 A > G/*ITGB3* rs3809865 A > T/*GP1BA* rs6065 C > T), the A.T.C combination, in which only *ITGB3* rs3809865 A > T changed to the T allele, was associated with increased RPL risk (OR = 1.485, *FDR-p* = 0.011). In the G.T.C combination, where *ITGB3* rs2317676 A > G changed from A to G, RPL risk significantly increased (OR = 28.333, *FDR-p* = 0.002). In the two-locus allele combination group (*ITGB3* rs3809865 A > T/*GP1BA* rs6065 C > T), the T.C combination was associated with increased RPL risk (OR = 1.514, *FDR-p* = 0.003).

Additionally, we performed haplotype analysis to confirm interactions between mutations within the same gene (*ITGB3, FGG, GP1BA,* and *PEAR1*; Table 3). The G.T combination, which contained the mutant alleles of *ITGB3* rs2317676 A > G and *ITGB3* rs3809865 A > T, and the A.G combination, where only *PEAR1* rs822442 C > A carried the mutant allele, were associated with increased RPL risk (OR = 26.910, *FDR-p* = 0.001; OR = 4.835, *FDR-p* = 0.003, respectively). No statistically significant combinations were identified in the haplotypes of *FGG* and *GP1BA*.

### 2.4. Genotype Combination Analysis of the ITGB3, FGG, GP1BA, PECAM1, and PEAR1 Gene Polymorphisms

To examine combinatorial effects among genotypes, we conducted a genotype combination analysis (Table 4, Appendix A). Prior to this assessment, MDR analysis was performed to reduce the dimensionality of the nine polymorphisms and identify meaningful genotype combinations. The following optimal MDR models were selected: *ITGB3* rs3809865 A > T/*FGG* rs1049636 T > C/*PECAM1* rs2812 C > T/*PEAR1* rs12137505 G > A (four-locus combination); *FGG* rs1049636 T > C/*PECAM1* rs2812 C > T/*PEAR1* rs12137505 G > A (three-locus combination); and *ITGB3* rs3809865 A > T/*GP1BA* rs6065 C > T (two-locus combination). The *FGG* rs1049636 TC/*PECAM1* rs2812 CT/*PEAR1* rs12137505 GG combination was consistently associated with decreased RPL risk in both the three-locus and four-locus genotype combinations (four-locus: AOR = 0.170, *p* = 0.013; three-locus: AOR = 0.341, *p* = 0.010). However, the *FDR-p* values were not statistically significant. In the two-locus genotype combination, the selected model was identical to the allele combination of *ITGB3* rs3809865/*GP1BA* rs6065. The *ITGB3* rs3809865 TT/*GP1BA* rs6065 CC genotype combination was associated with increased RPL risk and the *FDR-p* value was statistically significant (AOR = 3.107, *FDR-p* = 0.036). Among genotype combinations within the same gene, statistical significance was observed only for the two *PEAR1* polymorphisms (Table 4). The *PEAR1* rs822442 AA/*PEAR1* rs12137505 GA combination was associated with increased RPL risk (AOR = 12.661, *p* = 0.016), but the *FDR-p* value was not statistically significant (*FDR-p* = 0.112).

### 2.5. Variations in Clinical Parameters According to Polymorphism

RPL is a multifactorial condition influenced by genetic, endocrine, and environmental factors, which interact to collectively increase susceptibility. Genetic polymorphisms may affect clinical parameters and these variations could be linked to RPL prevalence. To determine whether specific polymorphisms influence particular clinical parameters, we evaluated genotype-based variations in clinical values using analysis of variance (Figure 2, Appendix A). The results indicated that immune cell proportions, specifically CD19+ B cells and CD56+ natural killer (NK) cells, were associated with the genotypes of *FGG* rs1049636 T > C and *PEAR1* rs12137505 G > A. The *FGG* rs1049636 TT genotype was associated with an increase in CD19+ B cell levels relative to the TC and CC genotypes (*p* = 0.034). Additionally, the proportion of CD56+ NK cells exhibited a sequential increase according to *PEAR1* rs12137505 G > A genotypes, following the order AA < AG < GG (*p* = 0.008). These findings suggest that increases in immune cell proportions promote autoimmune or inflammatory responses, thereby affecting pregnancy maintenance and potentially contributing to RPL. Our results confirmed that the *FGG* rs1049636 TT and *PEAR1* rs12137505 GG genotypes are associated with elevated immune cell levels and may be linked to RPL.

### 2.6. Synergistic Interactions Between Polymorphisms and Clinical Parameters

RPL is influenced by interactions between genetic polymorphisms and environmental factors, which collectively increase susceptibility. By analyzing interactions between polymorphisms in *ITGB3*, *FGG*, *GP1BA*, *PECAM1*, and *PEAR1* and environmental factors (clinical parameters), we identified the combined synergistic effects of genetic and clinical factors. For analysis, a TSH level ≥ 2.5 μU/mL was selected as the threshold, based on its association with pregnancy risk. The results demonstrated a synergistic effect between *ITGB3* rs3809865 A > T and TSH levels. Individuals with both *ITGB3* rs3809865 AT + TT genotypes and elevated TSH levels exhibited greater susceptibility to RPL relative to those with either the *ITGB3* rs3809865 AT + TT genotype alone or elevated TSH levels alone (AT + TT: AOR = 2.218; TSH ≥ 2.5 μU/mL: AOR = 1.346; AT + TT with TSH ≥ 2.5 μU/mL: AOR = 3.603; Figure 3A). Additionally, clinical values were divided into quartiles to assess variations in RPL risk across four groups. The results demonstrated that the synergistic effect between *ITGB3* rs3809865 A > T and TSH levels remained statistically significant. RPL risk sequentially increased according to TSH level (TSH < 1.083: AOR = 0.909; 1.083 ≤ TSH < 1.59: AOR = 2.567; 1.59 ≤ TSH < 2.39: AOR = 4.117; TSH ≥ 2.39: AOR = 6.817; Figure 3B). These findings suggest that the combination of genetic polymorphisms and clinical parameters contributes to RPL susceptibility.

Furthermore, to evaluate the predictive value of the nine SNPs and clinical factors for RPL, we performed a receiver operating characteristic (ROC) curve analysis of genetic and combined risk scores (Appendix A). The risk score combining the nine SNPs and PT levels reached an AUC of 0.826 (95% CI 0.791–0.858), the risk score combining the nine SNPs and aPTT reached an AUC of 0.752 (95% CI 0.713–0.788), and the risk score combining the nine SNPs, PT, and aPTT levels reached an AUC of 0.794 (95% CI 0.757–0.828).

## 3. Discussion

During pregnancy, the female body becomes prothrombotic due to changes in the coagulation system induced by female hormones, venous dilation, and blood flow restriction caused by uterine growth [21]. Additionally, inflammatory responses mediated by macrophages, dendritic cells, and NK cells are essential for embryo implantation and development, thus contributing to pregnancy maintenance [22]. Although thrombosis and inflammatory responses are critical for pregnancy, dysregulated responses can lead to pregnancy complications such as RPL, preeclampsia, and venous thrombosis [23,24,25]. Platelets play crucial roles in hemostasis and inflammatory responses [26]; they are integral to maternal and fetal health through their support of processes such as trophoblast implantation, placental formation, immune regulation, and thrombosis prevention [27,28,29]. Furthermore, platelet dysfunction has been associated with abnormal bleeding leading to menorrhagia, endometriosis, implantation failure, pregnancy loss, and platelet aggregation-induced ovarian failure [30,31,32,33,34].

Previous studies have identified genetic factors that may influence RPL, including several polymorphisms that are potentially associated with its prevalence [35]. Factor V Leiden (FVL) mutation (rs6025 G > A) and *MTHFR* (rs1801133 C > T, rs1801131A > C) gene mutations related to thrombosis have been reported in the context of RPL [36]. However, FVL mutations showed different distributions depending on ethnic group; Han et al. reported the first RPL case with FVL mutations [37], and *MTHFR* mutations were not associated with RPL prevalence in Koreans [38]. The findings in our prior study indicated that the platelet receptor *GP6* rs1654410 T > C and rs1654419 G > A polymorphisms were associated with decreased RPL risk [39]. Another study demonstrated that platelet counts during pregnancy were associated with *PEAR1* and *CBL* variants [40]. Therefore, the present study was conducted to identify genetic variants in platelet-associated genes that may contribute to RPL. Specifically, we investigated associations between polymorphisms in platelet receptors and ligands involved in platelet activation with RPL prevalence. Nine SNPs in genes related to platelet activation and coagulation (*ITGB3* rs2317676 A > G, rs3809865 A > T; *FGG* rs1049636 T > C, rs2066865 C > T; *GP1BA* rs2243093 T > C, rs6065 C > T; *PECAM1* rs2812 C > T; and *PEAR1* rs822442 C > A, rs12137505 G > A) were examined to determine their associations with RPL risk in a cohort of Korean women. To our knowledge, this study is the first to investigate the associations of *ITGB3* rs2317676 A > G, *ITGB3* rs3809865 A > T, *FGG* rs1049636 T > C, *GP1BA* rs2243093 T > C, *GP1BA* rs6065 C > T, *PECAM1* rs2812 C > T, *PEAR1* rs822442 C > A, and *PEAR1* rs12137505 G > A with RPL.

Genotype-based analysis revealed that the *ITGB3* rs3809865 TT genotype and the recessive model (AA + AT vs. TT) were significantly more frequent in RPL patients (*p* < 0.05), whereas the *FGG* rs1049636 TC genotype and the dominant model (TT vs. TC + CC) were significantly less frequent in patients with three or more pregnancy losses (*p* < 0.05). *ITGB3* encodes the β3 subunit of integrin aIIbβ3, which is one of the platelet surface receptors and is involved in platelet aggregation. It is also expressed together with the αV subunit and contributes to cell adhesion and migration [41]. Integrin αIIbβ3 is reportedly expressed in platelets, the extraplacental cone of placental cells, and the placental giant cells of the parietal yolk sac. Moreover, integrin αIIbβ3 regulates adhesion between placental cells and fibronectin during endometrial invasion. Glanzmann thrombasthenia, which is a hereditary platelet disorder, is characterized by defective platelet aggregation due to qualitative or quantitative abnormalities involving integrin αIIbβ3. Women with this condition have increased risks of menstrual bleeding, hemorrhagic ovarian cysts, endometrial hyperplasia, and endometrial polyps [42]. Additionally, *ITGB3* polymorphisms, including a novel heterozygous *ITGB3* p.T720del polymorphism [43], are reportedly associated with platelet aggregation dysfunction. In this study, we analyzed the *ITGB3* rs3809865 A > T variant, located in the 3′ UTR; we found that the rs3809865 TT genotype was associated with increased RPL risk. This variant has been reported to alter binding affinity for specific miRNAs, and it potentially modulates *ITGB3* expression levels [44]. *ITGB3* expression has been linked to the levels of *NOS3* and *SPP1*, which are genes involved in placental angiogenesis in sheep [45]. Notably, *ITGB3* levels were reduced at the embryo implantation site in a mouse model of polycystic ovary syndrome [46]. Among patients with RPL, *ITGB3* expression was significantly decreased in embryonic chorionic tissue, and the restoration of *ITGB3* levels via *H19* regulation enhanced trophoblast cell adhesion [47]. These findings suggest that *ITGB3* rs3809865 contributes to RPL development by influencing platelet function, embryo implantation, and stabilization.

Fibrinogen and fibrin stabilize the connection between the placenta and maternal attachment during fetal development [48]; they also play important roles in trophoblast cell proliferation [49]. There is evidence that women with hypofibrinogenemia or a fibrinogen variant (*FGA* rs6050A > G) exhibit a hypercoagulable state without thrombosis and have an increased risk of pregnancy loss [50]. Additionally, decreased fibrinogen levels have been closely associated with pregnancy-related complications, such as preeclampsia and placental abruption [51]. The rs1049636 T > C variant is a 3′ UTR polymorphism in the *FGG* gene. A previous study demonstrated that individuals with the TC/CC genotypes had significantly higher fibrinogen levels than those with the TT genotype at rs1049636 T > C [52]. Our investigation revealed that the rs1049636 TC genotype was associated with a decreased RPL risk. Another study indicated that the rs1049636 TC genotype was associated with a reduced risk of venous thromboembolism [53]. These findings suggest that the *FGG* rs1049636 T > C polymorphism alters fibrinogen levels, potentially influencing placental formation, embryo implantation, and thrombosis, thereby contributing to reduced RPL prevalence.

Clinical factors may also be associated with RPL development, and genetic polymorphisms may influence clinical parameters that contribute to this condition. In the present study, the *FGG* rs1049636 T > C and *PEAR1* rs12137505 G > A polymorphisms were associated with variations in immune cell counts (CD19+ B cells and CD56+ NK cells). Individuals with the *FGG* rs1049636 TT genotype exhibited significantly higher CD19+ B cell counts relative to those with the TC and CC genotypes. Additionally, individuals with the *PEAR1* rs12137505 GG genotype had significantly higher CD56+ NK cell counts relative to those with the GA and AA genotypes. Immune cells play a vital role in pregnancy; endometrial immune cells contribute to mechanisms of embryo implantation, survival, and development [54]. A previous study revealed an increased B cell count in the endometrium of patients with idiopathic RPL, as well as elevated NK cell counts in the uterus and peripheral blood [55]. Conversely, decreased NK cell and B cell counts were observed in low-risk pregnant women [56]. B cells and NK cells have also been implicated in pregnancy-related conditions, including preeclampsia, endometriosis, implantation failure, and preterm birth [57,58,59]. Furthermore, platelet receptor-related genetic mutations, CD16a expression, and antibody formation have been associated with immune cell alterations in the context of recurrent spontaneous abortion [60]. In individuals with the *FGG* rs1049636 TT and *PEAR1* rs12137505 GG genotypes, immune cell alterations may enhance RPL risk.

RPL is a heterogeneous condition resulting from interactions between genetic polymorphisms and environmental factors. In this study, the interaction between the *ITGB3* rs3809865 A > T polymorphism and TSH levels was associated with an increased RPL risk. TSH is a hormone secreted by the hypothalamus to regulate thyroid function; the thyroid gland plays critical roles in pregnancy maintenance and fetal development. Increased miscarriage risk has been detected in women with TSH levels between 2.5 and 4.87 mIU/L and those with levels ≥ 4.87 mIU/L relative to women with TSH levels between 0.4 and 2.5 mIU/L [61]. In our study, the presence of the *ITGB3* rs3809865 AT + TT genotype in combination with elevated TSH levels (≥2.5 mIU/L) was associated with an increased risk (up to 3.603-fold) of recurrent miscarriage. Additionally, as TSH levels increased, recurrent miscarriage risk sequentially increased to 2.567-fold, 4.117-fold, and 6.817-fold. These findings suggest an interaction between *ITGB3* and TSH levels during the development of recurrent miscarriage, indicating that the presence of both factors may contribute to RPL pathogenesis.

This study had some limitations. First, the mechanistic relationship between the observed polymorphisms and disease development is unclear; additional functional studies are required. Although the studied polymorphisms are located in coding and regulatory regions of the respective genes, their direct impacts on gene expression and regulatory mechanisms remain unclear. Further studies, including in vitro and in silico analyses, are needed to functionally validate the observed polymorphisms and their potential association with platelet function. Furthermore, although a potential synergistic effect between SNPs and TSH levels in terms of increasing RPL risk was identified, our findings remain preliminary. Further investigations concerning the direct effects of SNPs and TSH levels on RPL pathophysiology are necessary. Second, the sample size was limited due to difficulty in recruiting RPL patients. A larger cohort will be needed to confirm the observed associations. This study was exclusively conducted in Korean women, and associations with other ethnic populations have not been established. Therefore, the generalizability of the findings may be limited. Future analyses should include more diverse ethnic groups to enable these findings to be generalized to a broader global population. Third, this study selected variants based on previously reported findings and MAF, which may have limited the scope of the polymorphisms analyzed. Although access to large-scale genomic datasets such as the UK Biobank or GWAS results were not available, future studies incorporating such resources could help validate our findings and identify additional relevant variants.

## 4. Materials and Methods

### 4.1. Study Approval

All study protocols were reviewed and approved by the Institutional Review Board of CHA Bundang Medical Center in June 2011 (IRB number: 2010-01-123) and adhered to the tenets of the Declaration of Helsinki. Study participants were recruited from the South Korean provinces of Seoul and Gyeonggi-do between 2011 and 2024. Informed consent was obtained from all participants.

### 4.2. Study Population

Study participants were recruited between June 2011 and April 2023 at the Department of Obstetrics and Gynecology at the Infertility Medical Center, Bundang CHA Hospital. In total, 389 RPL patients and 375 controls were enrolled. Patients who experienced at least two consecutive pregnancy losses were classified as RPL patients. RPL was diagnosed after confirmation of pregnancy loss through measurement of human chorionic gonadotropin levels, ultrasound examination, and physical examination before the 20th week of pregnancy. This study focused on patients with unexplained RPL. Individuals with identifiable causes, including hormonal, genetic, anatomical, infectious, autoimmune, or thrombotic conditions, were excluded from the RPL group. Control participants were recruited from the CHA Bundang Medical Center and were required to have a normal 46,XX karyotype, regular menstrual cycles, a history of at least one natural pregnancy, and no history of pregnancy loss.

### 4.3. Estimation of Biochemical Factor Concentrations

Blood samples were collected from all participants in anticoagulant tubes after a 12 h fasting period. Plasma was separated by centrifugation at 1000× *g* for 15 min. Uric acid and total cholesterol levels were measured by enzymatic colorimetry (Roche Diagnostics, GmbH, Mannheim, Germany). Levels of high-density lipoprotein cholesterol were measured by enzymatic colorimetry using a set of commercial reagents (TBA 200FR NEO, Toshiba Medical Systems, Tochigi, Japan). Homocysteine levels were quantified by fluorescence polarization immunoassays using an Abbott IMx analyzer (Abbott Laboratories, Abbott Park, IL, USA). Folate and creatinine concentrations were determined using a competitive immunoassay with the ACS 180 Plus automated chemiluminescence system (Bayer Diagnostics, Tarrytown, NY, USA). Complete blood counts, including white blood cells, red blood cells, hemoglobin, and platelet counts, were obtained using the Sysmex XE 2100 automated hematology system (Sysmex Corporation, Kobe, Japan). Prothrombin time and activated partial thromboplastin time were measured using an automated photo-optical coagulometer (ACL TOP; Mitsubishi Chemical Medicence, Tokyo, Japan).

### 4.4. Flow Cytometry Analysis of Immune Cell Proportions

Immune cell proportions were analyzed by flow cytometry using CellQuest software version 5.1 (BD FACS Calibur; BD Biosciences, Franklin Lakes, NJ, USA). Peripheral blood mononuclear cells (2.5 × 10^5^) were stained for 30 min at 4 °C in the dark, washed twice with 2% phosphate-buffered saline containing 1% bovine serum albumin and 0.01% sodium azide (i.e., FACS wash buffer), and fixed with 1% formaldehyde (Sigma-Aldrich, St. Louis, MO, USA). Fluorescently labeled monoclonal antibodies (labeled with fluorescein isothiocyanate, phycoerythrin, peridinin chlorophyll protein, or allophycocyanin) specific for CD3, CD4, CD8, CD19, CD16, and CD56 were used at a dilution of 1:1000.

### 4.5. Hormone Assays

Female hormone levels (follicle-stimulating hormone, estradiol (E2), and luteinizing hormone) were measured on day 2 or 3 of the menstrual cycle when hormonal fluctuations are minimal and E2 interference is low to ensure a stable and accurate measurement of ovarian reserve and reproductive potential [62]. E2, thyroid-stimulating hormone (TSH), and prolactin levels were measured via radioimmunoassay (Beckman Coulter, Brea, CA, USA). Follicle-stimulating hormone and luteinizing hormone levels were measured by enzyme-linked immunosorbent assays (Siemens, Munich, Germany).

### 4.6. SNP Selection and Genetic Analysis

Five genes associated with platelet activation—*ITGB3*, *FGG*, *GP1BA*, *PECAM1*, and *PEAR1*—were selected for analysis. Polymorphisms in these genes were identified by reviewing studies that examined associations between thrombosis-related diseases (ischemic stroke, coronary artery disease, myocardial infarction, and venous thrombosis) and genetic variants [63,64,65,66,67,68,69,70]. In total, nine polymorphisms were included—*ITGB3* rs2317676 A > G, rs3809865 A > T; *FGG* rs1049636 T > C, rs2066865 T > C; *GP1BA* rs2243093 T > C, rs6065 C > T; *PECAM1* rs2812 C > T; and *PEAR1* rs822442 C > A, rs12137505 G > A. Among these, *FGG* rs1049636 and rs2066865 have been investigated in association with pregnancy loss and pregnancy state in at least one study [65,71], while *ITGB3* rs2317676 A > G, rs3809865 A > T; *GP1BA* rs2243093 T > C, rs6065 C > T; *PECAM1* rs2812 C > T; and *PEAR1* rs822442 C > A, rs12137505 G > A had not been studied in relation to pregnancy loss. Genomic DNA was extracted from anticoagulated peripheral blood using a G-DEX blood extraction kit (Intron, Seongnam, Republic of Korea). Genetic polymorphisms were analyzed by polymerase chain reaction–restriction fragment length polymorphism using restriction enzymes (Enzynomics, Daejeon, Republic of Korea) and by real-time polymerase chain reaction using TaqMan SNP genotyping kits (Applied Biosystems, Foster City, CA, USA). Information about primers and restriction enzymes utilized in the experiments is provided in Appendix A. To validate the results, DNA sequencing was performed on approximately 10–15% of randomly selected samples using an ABI 3730XL DNA Analyzer (Applied Biosystems). The concordance rate for quality control samples was 100%.

### 4.7. Statistical Analysis

To compare clinical characteristics between RPL patients and controls, the chi-square test was used for categorical data, whereas the independent *t*-test was utilized for continuous data. For variables that did not follow a normal distribution, the Mann–Whitney U test was applied. Logistic regression analysis was performed to evaluate associations of RPL prevalence with platelet-related gene polymorphisms; odds ratios (ORs) and 95% confidence intervals (CIs) were calculated. Adjusted odds ratios (AORs) for polymorphisms were determined using multiple logistic regression analysis, with age being regarded as a covariate. To control the false positive rate in the multiple comparison test, false discovery rate (FDR) correction was performed using the Benjamini–Hochberg method, while the false positive rate was set to no more than 5% [72]. One-way analysis of variance was conducted to examine differences in clinical factors according to genotype. The Kruskal–Wallis test was applied to variables that did not follow a normal distribution. Allele combinations for multiple loci were estimated using the expectation–maximization algorithm with HAPSTAT (version 3.0; University of North Carolina, Chapel Hill, NC, USA). In addition, ROC analysis was performed to evaluate the potential as predictors of SNPs and clinical factors. The area under the curve (AUC) indicates the discriminatory power of a diagnostic test. AUC 1.0 indicates 100% sensitivity and 100% specificity, while AUC 0.5 indicates a biomarker with non-discriminatory power. In general, an AUC value exceeding 0.75 indicates a significant biomarker with clinical potential. Analyses of interactions between genes were performed using the open-source multifactor dimensionality reduction (MDR) software package (v.2.0), available at www.epistasis.org (accessed on 12 May 2022). Statistical analyses were conducted using GraphPad Prism 4.0 (GraphPad Software Inc., San Diego, CA, USA) and MedCalc version 20.218 (MedCalc Software, Mariakerke, Belgium).

## 5. Conclusions

This study investigated associations of genetic polymorphisms in platelet activation-related genes (*ITGB3* rs2317676 A > G, rs3809865 A > T; *FGG* rs1049636 T > C, rs2066865 T > C; *GP1BA* rs2243093 T > C, rs6065 C > T; *PECAM1* rs2812 C > T; and *PEAR1* rs822442 C > A, rs12137505 G > A) with RPL prevalence. The *ITGB3* rs3809865 TT genotype was significantly associated with increased RPL risk, whereas the *FGG* rs1049636 TC genotype was significantly associated with a decreased risk of three or more pregnancy losses. Additionally, genotype and allele combinations were associated with RPL occurrence. One-way analysis of variance revealed that immune cell proportions (CD19+ and CD56+ cells) significantly differed according to *FGG* rs1049636 T > C and *PEAR1* rs12137505 G > A polymorphisms. Moreover, a synergistic effect between *ITGB3* rs3809865 A > T and TSH levels was observed, contributing to increased RPL risk. These findings indicate that *ITGB3* rs3809865 A > T and *FGG* rs1049636 T > C, located in regulatory regions, as well as *PEAR1* rs12137505 G > A, located in the coding sequence, may contribute to abnormal gene function and be associated with RPL prevalence. These discoveries could help identify novel prognostic biomarkers for RPL by incorporating *ITGB3* rs3809865 A > T, *FGG* rs1049636 T > C, and *PEAR1* rs12137505 G > A polymorphisms, along with other platelet activation- and coagulation-related gene polymorphisms and clinical factors such as immune cell counts and hormone levels.

## Figures and Tables

**Figure 1 ijms-26-07505-f001:**
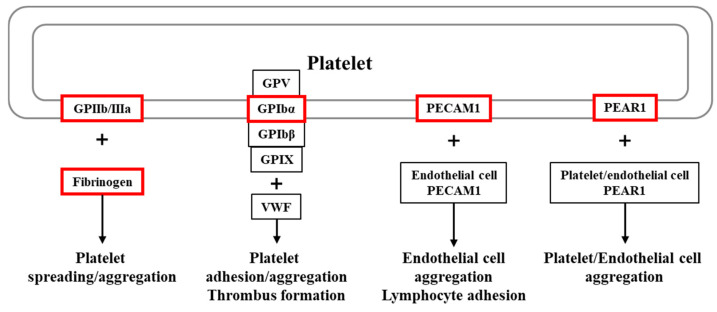
Graphical summary of platelet receptors and ligands investigated in this study.

**Figure 2 ijms-26-07505-f002:**
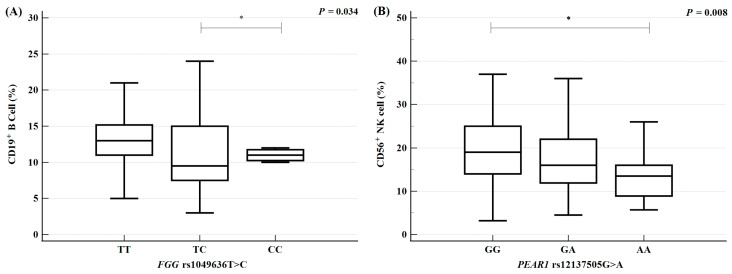
Association between differences in white blood cell levels; the proportions of CD19^+^ B and CD56^+^ NK cells; and the *FGG* rs1049636 T > C and *PEAR1* rs12137505 G > A polymorphisms in patients with recurrent pregnancy loss (RPL). (**A**) Association between CD19^+^ B cell proportions and the *FGG* rs1049636 T > C polymorphism. (**B**) Association between CD56^+^ NK cell proportions and the *PEAR1* rs12137505 G > A in patients with RPL.

**Figure 3 ijms-26-07505-f003:**
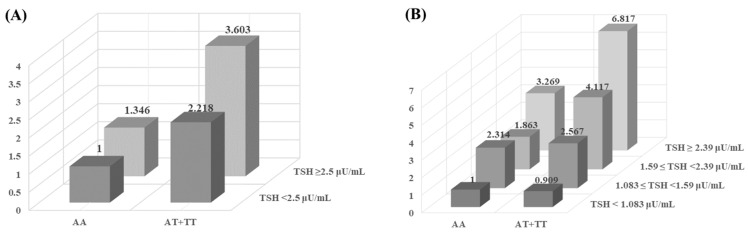
Synergic effect analysis for the interplay between TSH level and *ITGB3* rs3809865 A > T gene polymorphism in RPL prevalence. (**A**) Categorized TSH levels into two groups: < 2.5 μU/mL and ≥ 2.5 μU/mL. (**B**) Divided TSH levels into quartiles.

**Table 1 ijms-26-07505-t001:** Baseline characteristics between RPL patients and control subjects.

Characteristic	Controls (n = 375)	RPL Patients (n = 389)	*p*
Age	32.85 ± 4.18	33.42 ± 4.35	0.275 *
Previous pregnancy losses (n, %)	N/A	2.98 ± 1.48	N/A
Live births (n, %)	1.25 ± 0.03	N/A	N/A
Mean gestational age (weeks)	38.81 ± 1.44	7.44 ± 1.88	**<0.0001 ***
BMI	21.66 ± 3.08	21.42 ± 2.78	0.906 *
FSH (mIU/mL)	8.52 ± 4.43	7.90 ± 11.50	**<0.0001 ***
E2 (pg/mL)	29.71 ± 28.56	37.09 ± 29.77	**0.005 ***
LH (mIU/mL)	3.91 ± 4.57	6.37 ± 11.82	**<0.0001 ***
TSH (uIU/mL)	1.54 ± 1.12	2.13 ± 1.41	**<0.0001 ***
Prolactin (ng/mL)	11.33 ± 6.99	15.20 ± 11.85	0.123 *
AMH (ng/mL)	4.20 ± 2.89	3.64 ± 4.05	0.163 *
DHEA-S (ug/dL)	201.83 ± 72.45	148.40 ± 75.34	**0.007 ***
VEGF (pg/mL)	N/A	166.19 ± 130.03	N/A
PAI-1 (ng/mL)	N/A	11.22 ± 7.69	N/A
Anti-TPO (IU/mL)	11.54 ± 17.57	52.27 ± 250.60	**0.0001 ***
Thyroglobulin-Ab (IU/mL)	20.90 ± 18.17	62.00 ± 211.30	0.361 *
PT (sec)	10.60 ± 1.41	11.45 ± 1.66	**<0.0001 ***
aPTT (sec)	29.01 ± 3.39	32.25 ± 4.31	**<0.0001 ***
Protein C activity (%)	102.50 ± 24.75	98.55 ± 23.61	0.817
Protein S activity (%)	93.00 ± 46.67	69.40 ± 23.09	0.168
Homocysteine (umol/L)	7.56 ± 4.82	6.89 ± 1.96	0.914 *
Folate (ng/mL)	16.51 ± 12.86	15.62 ± 16.09	0.343 *
Hgb A1c (%)	5.41 ± 0.42	5.40 ± 0.32	0.813 *
Glucose (mg/dL)	96.56 ± 19.28	97.76 ± 16.67	**0.049 ***
BUN (mg/dL)	8.76 ± 2.77	10.39 ± 2.85	**<0.0001 ***
Creatinine (mg/dL)	0.60 ± 0.14	0.74 ± 0.12	**<0.0001 ***
Uric acid (mg/dL)	3.86 ± 0.97	3.83 ± 0.91	0.689 *
Total cholesterol (mg/dL)	232.87 ± 56.06	182.11 ± 46.82	**<0.0001 ***
Triglyceride (mg/dL)	224.10 ± 237.10	161.19 ± 137.74	**<0.0001 ***
LDL (mg/dL)	127.76 ± 40.59	109.57 ± 34.71	0.225 *
HDL (mg/dl)	74.62 ± 21.30	68.50 ± 19.05	0.427 *
WBC (10^3^/uL)	7.24 ± 2.48	6.93 ± 2.43	**0.038 ***
RBC (10^6^/uL)	4.13 ± 0.38	4.25 ± 0.63	**0.002 ***
Hgb (g/dL)	12.39 ± 1.93	12.68 ± 2.00	**0.0003 ***
Hct (%)	36.55 ± 3.19	37.57 ± 3.52	**<0.0001 ***
MCV (fL)	88.13 ± 6.71	88.97 ± 6.69	0.107 *
MCH (pg)	30.64 ± 14.83	29.92 ± 2.40	0.480 *
MCHC (g/dL)	33.68 ± 1.20	33.54 ± 1.42	0.307 *
RDW (%)	13.35 ± 1.20	13.12 ± 1.34	**<0.0001 ***
PLT (10^3^/ul)	236.47 ± 60.12	253.62 ± 59.25	**0.0001 ***
PDW (fL)	12.57 ± 2.43	15.59 ± 10.21	**0.0001 ***
MPV (fL)	9.06 ± 4.62	8.51 ± 5.06	**0.001 ***
Seg (%)	70.23 ± 8.53	62.65 ± 11.98	**<0.0001 ***
Lym (%)	21.40 ± 7.18	28.44 ± 10.60	**<0.0001 ***
Mono (%)	5.29 ± 1.66	5.64 ± 2.69	0.272 *
Eo (%)	1.59 ± 1.30	2.12 ± 1.76	**0.0002 ***
Baso (%)	0.35 ± 0.24	0.44 ± 0.31	**0.001 ***
CD56 NK cell (%)	N/A	17.37 ± 7.79	N/A
CD3 (pan T) (%)	N/A	67.01 ± 0.89	N/A
CD4 (helper T) (%)	N/A	36.39 ± 7.41	N/A
CD8 (suppressor T) (%)	N/A	27.33 ± 0.79	N/A
CD19 (B-cell) (%)	N/A	12.66 ± 4.88	N/A

Note—RPL: recurrent pregnancy loss; BMI: body mass index; FSH: follicle-stimulating hormone; E2: estradiol; LH: luteinizing hormone; TSH: thyroid-stimulating hormone; AMH: anti-Mullerian hormone; DHEA-S: dehydroepiandrosterone-sulfate; VEGF: vascular endothelial growth factor; PAI-1: plasminogen activator inhibitor-1; PT: prothrombin time; aPTT: activated partial thromboplastin time; Hgb A1c: hemoglobin A1c; BUN: blood urea nitrogen; HDL: high-density lipoprotein; LDL: low-density lipoprotein; WBC: white blood cell; RBC: red blood cell; Hgb: hemoglobin; Hct: hematocrit; MCV: mean corpuscular volume; MCH: mean corpuscular hemoglobin; MCHC: mean corpuscular hemoglobin concentration; RDW: red blood cell distribution width; PLT: platelet; PDW: platelet distribution width; MPV: mean platelet component; Seg: segmented neutrophils; Lym: lymphocytes; Mono: monocytes; Eo: eosinophils; Baso: basophils. *p* was calculated using the independent sample *t*-test and Mann–Whitney test for continuous variables. * Mann–Whitney test.

**Table 2 ijms-26-07505-t002:** Comparison of genotype frequencies of the *ITGB3* rs3809865 A > T and *FGG* rs1049636 T *>* C polymorphisms between the RPL and control subjects.

Genotypes	Controls (n = 375)	RPL (n = 389)	AOR (95% CI)	*p*	*FDR-p*	PL ≥ 3 (n = 205)	AOR (95% CI)	*p*	*FDR-p*	PL ≥ 4 (n = 76)	AOR (95% CI)	*p*	FDR*-p*
*ITGB3* rs3809865A > T													
AA	233 (62.1)	218 (56.0)	1.000 (reference)			112 (54.6)	1.000 (reference)			40 (52.6)	1.000 (reference)		
AT	129 (34.4)	142 (36.5)	1.170 (0.865–1.584)	0.309	0.745	73 (35.6)	1.158 (0.802–1.672)	0.435	0.670	28 (36.8)	1.248 (0.735–2.120)	0.413	0.990
TT	13 (3.5)	29 (7.5)	**2.505 (1.262–4.969)**	**0.009**	0.081	20 (9.8)	**3.255 (1.551–6.830)**	**0.002**	**0.018**	8 (10.5)	**3.613 (1.403–9.307)**	**0.008**	0.072
Dominant (AA vs. AT + TT)			1.285 (0.962–1.718)	0.090	0.288		1.345 (0.950–1.903)	0.094	0.306		1.461 (0.889–2.402)	0.135	0.608
Recessive (AA + AT vs. TT)			**2.302 (1.176–4.508)**	**0.015**	0.135		**3.050 (1.479–6.292)**	**0.0** **03**	**0.0** **27**		**3.301 (1.316–8.280)**	**0.0** **11**	0.099
HWE-*p*	0.342	0.382											
*FGG* rs1049636T > C													
TT	234 (62.4)	265 (68.1)	1.000 (reference)			145 (70.7)	1.000 (reference)			56 (73.7)	1.000 (reference)		
TC	127 (33.9)	107 (27.5)	0.738 (0.540–1.009)	0.057	0.513	54 (26.3)	**0.673 (0.460–0.987)**	**0.043**	0.387	17 (22.4)	**0.556 (0.310–0.997)**	**0.049**	0.441
CC	14 (3.7)	17 (4.4)	1.065 (0.513–2.210)	0.866	0.926	6 (2.9)	0.650 (0.243–1.740)	0.391	0.790	3 (3.9)	0.880 (0.244–3.172)	0.845	0.845
Dominant (TT vs. TC + CC)			0.771 (0.571–1.040)	0.088	0.288		**0.669 (0.463–0.968)**	**0.033**	0.297		0.588 (0.339–1.022)	0.060	0.540
Recessive (TT + TC vs. CC)			1.173 (0.569–2.417)	0.666	0.797		0.734 (0.276–1.950)	0.534	0.939		1.046 (0.293–3.737)	0.945	0.945
HWE-*p*	0.525	0.149											

RPL: recurrent pregnancy loss; COR: crude odds ratio; AOR: adjusted odds ratio; FDR: false discovery rate; HWE: Hardy–Weinberg equilibrium, adjusted by age.

**Table 3 ijms-26-07505-t003:** Allele combination analysis of *ITGB3*, *FGG*, *GP1BA*, *PECAM1*, and *PEAR1* gene polymorphisms in RPL and control subjects.

Allele Combination	Controls (2n = 750)	RPL (2n = 778)	OR (95% CI)	*p*	*FDR-p*
*ITGB3* rs2317676 A > G/*ITGB3* rs3809865 A > T/*FGG* rs1049636 T > C/*FGG* rs2066865 T > C/*GP1BA* rs2243093 T > C/*GP1BA* rs6065 C > T/*PECAM1* rs2812 C > T/*PEAR1* rs822442 C > A/*PEAR1* rs12137505 G > A					
A.A.T.T.T.C.C.C.G	43 (5.7)	55 (7.1)	1.000 (reference)		
A.A.T.T.T.C.T.A.A	17 (2.3)	0 (0.0)	**0.022 (0.001–0.383)**	**0.0001**	**0.008**
A.A.T.C.C.C.C.A.A	30 (4.0)	4 (0.6)	**0.104 (0.034–0.319)**	**0.0001**	**0.008**
A.T.T.T.T.C.C.C.G	36 (4.8)	11 (1.4)	**0.239 (0.109–0.524)**	**0.0002**	**0.010**
A.T.T.C.T.C.C.C.G	0 (0.0)	17 (2.2)	**27.430 (1.603–469.400)**	**0.001**	**0.023**
A.T.C.C.C.C.C.C.G	9 (1.1)	0 (0.0)	**0.041 (0.002–0.729)**	**0.001**	**0.030**
*ITGB3* rs2317676 A > G/*ITGB3* rs3809865 A > T/*FGG* rs1049636 T > C/*PEAR1* rs12137505 G > A					
A.A.T.G	196 (26.2)	209 (26.8)	1.000 (reference)		
A.T.C.G	29 (3.9)	12 (1.5)	**0.388 (0.193–0.782)**	**0.006**	**0.045**
G.T.C.A	0 (0.0)	9 (1.2)	**17.820 (1.030–308.400)**	**0.004**	**0.045**
*ITGB3* rs2317676 A > G/*ITGB3* rs3809865 A > T/*GP1BA* rs6065 C > T					
A.A.C	433 (57.7)	382 (49.1)	1.000 (reference)		
A.A.T	32 (4.3)	51 (6.5)	**1.807 (1.137–2.870)**	**0.011**	**0.023**
A.T.C	129 (17.2)	169 (21.8)	**1.485 (1.137–1.940)**	**0.004**	**0.011**
G.T.C	0 (0.0)	12 (1.6)	**28.330 (1.671–480.500)**	**0.0003**	**0.002**
*ITGB3* rs3809865 A > T/*GP1BA* rs6065 C > T					
A.C	553 (73.71)	518 (66.64)	1.000 (reference)		
A.T	42 (5.62)	60 (7.65)	**1.525 (1.010–2.303)**	**0.044**	0.066
T.C	129 (17.22)	183 (23.46)	**1.514 (1.173–1.955)**	**0.001**	**0.003**
*ITGB3* rs2317676 A > G/*ITGB3* rs3809865 A > T					
A.A	465 (62.0)	432 (55.6)	1.000 (reference)		
A.T	155 (20.7)	188 (24.1)	**1.306 (1.017–1.676)**	**0.036**	0.054
G.A	130 (17.3)	146 (18.7)	1.209 (0.923–1.584)	0.169	0.169
G.T	0 (0.0)	12 (1.6)	**26.910 (1.587–456.200)**	**0.0004**	**0.001**
*FGG* rs1049636 T > C/*FGG* rs2066865 T > C					
T.T	384 (51.1)	384 (49.4)	1.000 (reference)		
T.C	212 (28.2)	253 (32.5)	5.486 (0.276–109.100)	0.259	0.267
C.T	2 (0.2)	9 (1.1)	**4.500 (0.966–20.970)**	**0.036**	0.108
C.C	154 (20.5)	132 (17.0)	0.857 (0.653–1.125)	0.267	0.267
*GP1BA* rs2243093 T > C/*GP1BA* rs6065 C > T					
T.C	489 (65.3)	511 (65.7)	1.000 (reference)		
T.T	64 (8.5)	71 (9.2)	1.062 (0.741–1.522)	0.784	0.784
C.C	193 (25.7)	190 (24.5)	0.942 (0.744–1.192)	0.62	0.784
C.T	4 (0.6)	6 (0.7)	1.435 (0.403–5.119)	0.753	0.784
*PEAR1* rs822442 C > A/*PEAR1* rs12137505 G > A					
C.G	419 (55.91)	416 (53.46)	1.000 (reference)		
C.A	105 (13.95)	94 (12.09)	0.902 (0.662–1.229)	0.512	0.512
A.G	5 (0.62)	24 (3.09)	**4.835 (1.827–12.800)**	**0.001**	**0.003**
A.A	221 (29.51)	244 (31.36)	1.112 (0.886–1.395)	0.359	0.512

RPL: recurrent pregnancy loss; OR: odds ratio; CI: confidence interval.

**Table 4 ijms-26-07505-t004:** Genotype combination of *ITGB3*, *FGG*, *GP1BA*, *PECAM1*, and *PEAR1* gene polymorphisms in RPL and control subjects.

Genotype Combination	Controls (n = 375)	RPL Patients (n = 389)	AOR (95% CI)	*p*	*FDR-p*
*ITGB3* rs3809865 A > T/*FGG* rs1049636 T > C/*PECAM1* rs2812 C > T/*PEAR1* rs12137505 G > A					
AA/TT/CC/GG	19 (5.1)	28 (7.2)	1.000 (Reference)		
AA/TT/CT/GG	22 (5.9)	11 (2.8)	**0.350 (0.137–0.893)**	**0.028**	0.581
AT/TC/CT/GG	12 (3.2)	3 (0.8)	**0.170 (0.042–0.683)**	**0.013**	0.580
AT/TC/CT/GA	15 (4.0)	7 (1.8)	**0.319 (0.109–0.938)**	**0.038**	0.581
*FGG* rs1049636 T > C/*PECAM1* rs2812 C > T/*PEAR1* rs12137505 G > A					
TT/CC/GG	27 (7.2)	46 (11.8)	1.000 (Reference)		
TT/CC/GA	67 (17.9)	55 (14.1)	**0.485 (0.267–0.880)**	**0.017**	0.130
TC/CT/GG	24 (6.4)	14 (3.6)	**0.341 (0.151–0.769)**	**0.010**	0.130
TC/CT/GA	32 (8.5)	22 (5.7)	**0.404 (0.196–0.833)**	**0.014**	0.130
*ITGB3* rs3809865 A > T/*GP1BA* rs6065 C > T					
AA/CC	202 (53.9)	174 (44.7)	1.000 (reference)		
TT/CC	9 (2.4)	22 (5.7)	**3.107 (1.382–6.987)**	**0.006**	**0.036**
*PEAR1* rs822442 C > A/*PEAR1* rs12137505 G > A					
CC/GG	115 (30.7)	110 (28.3)	1.000 (reference)		
AA/GA	1 (0.3)	12 (3.1)	**12.661 (1.618–99.085)**	**0.016**	0.112

RPL: recurrent pregnancy loss; AOR: adjusted odds ratio, adjusted by age. *p*-value was calculated using logistic regression.

## Data Availability

The original contributions presented in this study are included in the article. Further inquiries can be directed to the corresponding authors.

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
