# Peer review of "Genetic Associations of ITGB3, FGG, GP1BA, PECAM1, and PEAR1 Polymorphisms and the Platelet Activation Pathway with Recurrent Pregnancy Loss in the Korean Population"

_ijms, 2025, doi:10.3390/ijms26157505_

Round 1

Reviewer 1 Report

Comments and Suggestions for Authors

The present study addresses an important topic, revealing interesting findings regarding potential associations between platelet-related gene SNPs with RPL prevalence.  Careful proofreading is needed to improve language quality and eliminate errors (e.g., page 1, line 40 : "increased risk" must be replaced with "decreased risk" - potential typo error ). 

Addressing these minor points will improve the manuscript’s clarity and scientific rigor.

Comments on the Quality of English Language

The overall quality of the English language in the manuscript is adequate but would benefit from careful proofreading to improve clarity and readability. I noticed a typo error in the "Abstract" section in page 1, line 40, where "increased risk" must be replaced with "decreases risk".

Author Response

The present study addresses an important topic, revealing interesting findings regarding potential associations between platelet-related gene SNPs with RPL prevalence.  Careful proofreading is needed to improve language quality and eliminate errors (e.g., page 1, line 40 : "increased risk" must be replaced with "decreased risk" - potential typo error). 

Addressing these minor points will improve the manuscript’s clarity and scientific rigor.

=> Thank you for critical comments. We tried to augment some contents as you suggested. We corrected the English before submission in BioScienceWriters. And, we have thoroughly checked the revised manuscript.

The overall quality of the English language in the manuscript is adequate but would benefit from careful proofreading to improve clarity and readability. I noticed a typo error in the "Abstract" section in page 1, line 40, where "increased risk" must be replaced with "decreases risk".

=> Thank you for your comment. So, we revised the sentence.

“The FGG rs1049636 T>C polymorphism was associated with a decreased risk in women who had three or more pregnancy losses.” [Abstract section, page1, line 40]

Reviewer 2 Report

Comments and Suggestions for Authors

The manuscript investigates associations between platelet activation-related gene polymorphisms and recurrent pregnancy loss (RPL) in a Korean cohort. The authors analyze nine SNPs in five genes (ITGB3, FGG, GP1BA, PECAM1, PEAR1) in 389 RPL patients and 375 controls, integrating genotyping with clinical and immunologic parameters. This topic is clinically important, given the high burden of unexplained RPL and the potential role of thrombosis and immune dysregulation. The study is methodologically solid and reasonably well presented, with thoughtful integration of genetic and clinical data. However, several key concerns limit the current impact and interpretability of the findings.

1.Variant selection strategy needs clarification and expansion. The choice of polymorphisms is limited and seems outdated in the context of currently available resources like the UK Biobank. While the authors selected SNPs based on previous clinical studies and MAF, the rationale could be strengthened by referencing genome-wide association studies (GWAS) or more comprehensive datasets.
Recommendation:  Explicitly acknowledge the limitation in the manuscript. Clearly state that the polymorphisms selected are hypothesis-driven and based on previous studies with potentially limited coverage. Mention explicitly in the discussion section that larger genomic datasets (such as the UK Biobank) might offer a more comprehensive variant set. If such data were inaccessible, acknowledge this limitation and suggest this as a future direction.

2.Table 1 presentation needs optimization. Table 1 is very large and includes both significant and non-significant comparisons. This reduces readability and distracts from the key findings.
Recommendation: Present only statistically significant comparisons in the main manuscript table. Move non-significant clinical characteristics to a supplementary table to improve clarity and focus.

3.Overloaded and partially inaccurate statistical methods section. The statistical methods section lists multiple tests (chi-square, t-test, Mann–Whitney U, ANOVA, Kruskal–Wallis, logistic regression, FDR, MDR, HAPSTAT, ROC), and it is unclear whether all were genuinely used. Moreover, 'sex' is listed as a covariate in a female-only cohort, which is a clear error.
Recommendation: Revise the statistical methods to include only those methods actually used and justify the choice of each. Remove “sex” as a covariate. Clarify when t-test vs. Mann–Whitney and ANOVA vs. Kruskal–Wallis were used based on distribution.

4.Absence of ROC/AUC analysis weakens clinical interpretability. Although the study identifies significant associations and allele/genotype combinations, there is no ROC or AUC analysis to evaluate their predictive power. This omission limits the clinical utility and translational potential of the results.
Recommendation: Include ROC curve analysis and report AUC values for key SNPs and combinations (especially ITGB3 rs3809865 and FGG rs1049636). This would allow readers to assess diagnostic performance more intuitively.

This is a valuable and clinically relevant study that could make a strong contribution to the field of reproductive genetics. However, it would benefit substantially from clearer justification of variant selection, addition of ROC/AUC analysis, optimization of table presentation, and a careful revision of the statistical methods section. With these improvements, the manuscript would be significantly strengthened. After this major revision, another peer review will be required before the manuscript can be published.

Author Response

The manuscript investigates associations between platelet activation-related gene polymorphisms and recurrent pregnancy loss (RPL) in a Korean cohort. The authors analyze nine SNPs in five genes (ITGB3, FGG, GP1BA, PECAM1, PEAR1) in 389 RPL patients and 375 controls, integrating genotyping with clinical and immunologic parameters. This topic is clinically important, given the high burden of unexplained RPL and the potential role of thrombosis and immune dysregulation. The study is methodologically solid and reasonably well presented, with thoughtful integration of genetic and clinical data. However, several key concerns limit the current impact and interpretability of the findings.

=> Thank you for critical comments. We tried to augment some contents as you suggested. We revised our insufficient descriptions as follows:

  1. Variant selection strategy needs clarification and expansion. The choice of polymorphisms is limited and seems outdated in the context of currently available resources like the UK Biobank. While the authors selected SNPs based on previous clinical studies and MAF, the rationale could be strengthened by referencing genome-wide association studies (GWAS) or more comprehensive datasets.
    Recommendation: Explicitly acknowledge the limitation in the manuscript. Clearly state that the polymorphisms selected are hypothesis-driven and based on previous studies with potentially limited coverage. Mention explicitly in the discussion section that larger genomic datasets (such as the UK Biobank) might offer a more comprehensive variant set. If such data were inaccessible, acknowledge this limitation and suggest this as a future direction.

=> Thank you for your comment. We agree with your comments. So, we added the following paragraph to the discussion section.

“Third, this study selected variants based on previously reported findings and MAF, which may have limited the scope of polymorphisms analyzed. Although access to large-scale genomic datasets such as the UK Biobank or GWAS results were not available, future studies incorporating such resources could help validate our findings and identify additional relevant variants.” [Discussion section, page12-13, line 425-430]

  1. Table 1 presentation needs optimization. Table 1 is very large and includes both significant and non-significant comparisons. This reduces readability and distracts from the key findings.
    Recommendation: Present only statistically significant comparisons in the main manuscript table. Move non-significant clinical characteristics to a supplementary table to improve clarity and focus.

=> Thank you for your comment. We agree with your comments. So, we revised Table 2 to the Result section and added Supplementary Table 2 to the supplementary material.

  1. Overloaded and partially inaccurate statistical methods section. The statistical methods section lists multiple tests (chi-square, t-test, Mann–Whitney U, ANOVA, Kruskal–Wallis, logistic regression, FDR, MDR, HAPSTAT, ROC), and it is unclear whether all were genuinely used. Moreover, 'sex' is listed as a covariate in a female-only cohort, which is a clear error.
    Recommendation: Revise the statistical methods to include only those methods actually used and justify the choice of each. Remove “sex” as a covariate. Clarify when t-test vs. Mann–Whitney and ANOVA vs. Kruskal–Wallis were used based on distribution.

=> Thank you for your comment. We agree with your comments. So, we revised sentences to the material and method section.

“For variables that did not follow a normal distribution, the Mann–Whitney U test was applied.” [Material and Method section, page14, line 507-508]

“Adjusted odds ratios (AORs) for polymorphisms were determined using multiple logistic regression analysis, with age regarded as a covariate.” [Material and Method section, page14, line 510-512]

“And Kruskal–Wallis was applied to variables that did not follow a normal distribution.” [Material and Method section, page14, line 515-516]

  1. Absence of ROC/AUC analysis weakens clinical interpretability. Although the study identifies significant associations and allele/genotype combinations, there is no ROC or AUC analysis to evaluate their predictive power. This omission limits the clinical utility and translational potential of the results.
    Recommendation: Include ROC curve analysis and report AUC values for key SNPs and combinations (especially ITGB3 rs3809865 and FGG rs1049636). This would allow readers to assess diagnostic performance more intuitively.

=> Thank you for your comment. We agree with your comments. So, we added the following paragraph to the result and method section.

“Furthermore, to evaluate the predictive value of 9 SNPs and clinical factors for RPL, we performed a receiver operating characteristic (ROC) curve analysis of genetic, and combined risk scores (Supplementary Figure 1). The risk score combining the nine SNPs and PT levels reached an AUC of 0.826 (95% CI 0.791–0.858), the risk score combining the nine SNPs and aPTT reached an AUC of 0.752 (95% CI 0.713–0.788), and the risk score combining the nine SNPs, PT, and aPTT levels reached an AUC of 0.794 (95% CI 0.757–0.828).” [Result ection, page10, line 292-299]

“In addition, ROC analysis was performed to evaluate the potential as predictors of SNPs and clinical factors. The area under the curve (AUC) indicates the discriminatory power of a diagnostic test. AUC 1.0 indicates 100% sensitivity and 100% specificity, while AUC 0.5 indicates a biomarker with non-discriminatory power. In general, an AUC value exceeding 0.75 indicates a significant biomarker with clinical potential.” [Material and Method section, page14, line 518-522]

Reviewer 3 Report

Comments and Suggestions for Authors

The manuscript by Ko et al. investigates the genetic association between selected SNPs in genes related to platelet function (ITGB3, FGG, GP1BA, PECAM1, PEAR1) and the risk of recurrent pregnancy loss (RPL) in a Korean female population. Utilizing a case-control design, the authors assessed 389 RPL patients and 375 healthy controls for polymorphisms and analyzed correlations with clinical variables relevant to reproductive health. The study addresses an important and under-researched area in reproductive genetics, focusing on specific genes involved in platelet function and their association with RPL.
The results are potentially impactful for the identification of genetic biomarkers and the development of future therapeutic strategies for RPL.

The manuscript contains several long paragraphs that could benefit from clearer subheadings and more concise language for easier reading.

Some sections (such as the introduction and background on platelet biology) could be condensed to improve focus on the genetic association aspect.

The discussion of study limitations is brief. Additional emphasis should be given not only to the restriction to Korean ethnicity, but also to the lack of functional validation of the SNPs, and possible selection bias.
Moreover, the rationale for the inclusion of some polymorphisms (e.g., why these and not others in the same genes or in other genes) could be explained in more detail.

Functional studies or in-silico modeling to support a mechanistic link between the identified SNPs and altered platelet function in RPL would strengthen the conclusions.

Review the manuscript for minor typographical errors and inconsistencies, particularly in gene/SNP nomenclature and reference formatting (in line 319, 521, 522 for example).

In line 419 the sentence about FMR1 gene is not clear.

Author Response

The manuscript by Ko et al. investigates the genetic association between selected SNPs in genes related to platelet function (ITGB3, FGG, GP1BA, PECAM1, PEAR1) and the risk of recurrent pregnancy loss (RPL) in a Korean female population. Utilizing a case-control design, the authors assessed 389 RPL patients and 375 healthy controls for polymorphisms and analyzed correlations with clinical variables relevant to reproductive health. The study addresses an important and under-researched area in reproductive genetics, focusing on specific genes involved in platelet function and their association with RPL.
The results are potentially impactful for the identification of genetic biomarkers and the development of future therapeutic strategies for RPL.

=> Thank you for critical comments. We tried to augment some contents as you suggested. We revised our insufficient descriptions as follows:

1) The manuscript contains several long paragraphs that could benefit from clearer subheadings and more concise language for easier reading.

=> Thank you for your comment. We agree with your comments. So, we have revised the long sentence to improve clarity and brevity.

2) Some sections (such as the introduction and background on platelet biology) could be condensed to improve focus on the genetic association aspect.

=> Thank you for your comment. We agree with your comments. So, we have deleted some sentence to the introduction section.

3) The discussion of study limitations is brief. Additional emphasis should be given not only to the restriction to Korean ethnicity, but also to the lack of functional validation of the SNPs, and possible selection bias. Moreover, the rationale for the inclusion of some polymorphisms (e.g., why these and not others in the same genes or in other genes) could be explained in more detail.

=> Thank you for your comment. We agree with your comments. So, we added the following paragraph to the discussion section.

“Further studies, including in vitro and in silico analyses, are needed to functionally validate the observed polymorphisms and their potential association with platelet function.” [Discussion section, page12, line 415-417]

“Therefore, the generalizability of the findings may be limited.” [Discussion section, page12, line 423-424]

“Third, this study selected variants based on previously reported findings and MAF, which may have limited the scope of polymorphisms analyzed. Although access to large-scale genomic datasets such as the UK Biobank or GWAS results was not available, future studies incorporating such resources could help validate our findings and identify additional relevant variants.” [Discussion section, page12-13, line 425-430]

4) Functional studies or in-silico modeling to support a mechanistic link between the identified SNPs and altered platelet function in RPL would strengthen the conclusions.

=> Thank you for your comment. We agree with your comments. So, we added the following paragraph to the discussion section.

“Further studies, including in vitro and in silico analyses, are needed to functionally validate the observed polymorphisms and their potential association with platelet function.” [Discussion section, page12, line 415-417]

5) Review the manuscript for minor typographical errors and inconsistencies, particularly in gene/SNP nomenclature and reference formatting (in line 319, 521, 522 for example).

=> Thank you for your comment. We agree with your comments. So, the SNPs were consistently presented in the format of gene name + rs number + nucleotide change (e.g., A>G), and the genotypes were presented as gene name + rs number + genotype throughout the manuscript.

=> Thank you for your comment. We agree with your comments. So, we revised the 2, 4 reference formatting. And the fourth reference has been revised according to the citation format for books.

“2. Lee, S.Y. Reproductive health status and policy. Health Welf Policy Forum 2022, 308, 94–104, doi:10.23062/2022.06.8.” [Reference section, page16, line 585]

“4. RCOG. Green-Top Guideline No. 17 the Investigation and Treatment of Couples with Recurrent First-Trimester and Second-Trimester Miscarriage; RCOG: London, UK, 2011.” [Reference section, page16, line 588-589]

6) In line 419 the sentence about FMR1 gene is not clear.

=> Thank you for your comment. We agree with your comments. So, we considered that the sentence was not closely related to the main content of the manuscript and therefore decided to remove it.

Round 2

Reviewer 2 Report

Comments and Suggestions for Authors

The authors have revised the article, which I now recommend for publication.